# The Validity and Reliability of Self-Reported Adherence to Using Offloading Treatment in People with Diabetes-Related Foot Ulcers

**DOI:** 10.3390/s23094423

**Published:** 2023-04-30

**Authors:** Anas Ababneh, Kathleen Finlayson, Helen Edwards, David G. Armstrong, Bijan Najafi, Jaap J. van Netten, Peter A. Lazzarini

**Affiliations:** 1Faculty of Nursing, Applied Science Private University, Amman 11931, Jordan; 2School of Nursing, Queensland University of Technology, Brisbane, QLD 4059, Australia; 3Centre for Healthcare Transformation, Queensland University of Technology, Brisbane, QLD 4059, Australia; 4Keck School of Medicine, University of Southern California, Los Angeles, CA 90089, USA; 5Interdisciplinary Consortium on Advanced Motion Performance (iCAMP), Michael E. DeBakey Department of Surgery, Baylor College of Medicine, Houston, TX 77030, USA; 6Department of Rehabilitation Medicine, 1105 AZ Amsterdam UMC, University of Amsterdam, Meibergdreef 9, 1012 WX Amsterdam, The Netherlands; 7Amsterdam Movement Sciences, Ageing & Vitality and Rehabilitation and Development, 1105 AZ Amsterdam, The Netherlands; 8School of Public Health and Social Work, Queensland University of Technology, Brisbane, QLD 4059, Australia; 9Allied Health Research Collaborative, The Prince Charles Hospital, Brisbane, QLD 4032, Australia

**Keywords:** adherence, diabetic foot, foot ulcer, offloading, self-report

## Abstract

Adherence to using offloading treatment is crucial to healing diabetes-related foot ulcers (DFUs). Offloading adherence is recommended to be measured using objective monitors. However, self-reported adherence is commonly used and has unknown validity and reliability. This study aimed to assess the validity and reliability of self-reported adherence to using removable cast walker (RCW) offloading treatment among people with DFUs. Fifty-three participants with DFUs using RCWs were included. Each participant self-reported their percentage adherence to using their RCW of total daily steps. Participants also had adherence objectively measured using dual activity monitors. After one week, a subset of 19 participants again self-reported their percentage adherence to investigate test–retest reliability. Validity was tested using Pearson’s r and Bland–Altman tests, and reliability using Cohen’s kappa. Median (IQR) self-reported adherence was greater than objectively measured adherence (90% (60–100) vs. 35% (19–47), *p* < 0.01). There was fair agreement (r = 0.46; *p* < 0.01) and large 95% limits of agreement with significant proportional bias (β = 0.46, *p* < 0.01) for validity, and minimal agreement for test–retest reliability (K = 0.36; *p* < 0.01). The validity and reliability of self-reported offloading adherence in people with DFU are fair at best. People with DFU significantly overestimate their offloading adherence. Clinicians and researchers should instead use objective adherence measures.

## 1. Introduction

Diabetes-related foot ulcers (DFUs) affect around 20 million people globally [1,2] and are responsible for high incidences of hospitalisations, amputations, disability, treatment costs, and premature death [1,3,4,5,6]. DFUs are typically caused by high plantar pressure in people with a loss of foot sensation from diabetes-related peripheral neuropathy [2].

International DFU guidelines have long concluded that offloading high plantar pressure from DFUs is crucial to healing people with DFU [7]. Non-removable offloading devices are recommended as the gold standard offloading treatment for people with DFU [8,9]. However, implementing non-removable offloading devices in clinical practice has been found to be challenging due to the technical skills required to customise non-removable offloading treatments, such as total contact casts [10,11,12,13], and people’s preferences for not using offloading devices all the time, such as when not bearing weight [10,12]. The second-choice recommended offloading treatments are removable offloading devices, such as removable cast walkers, which are more commonly used in clinical practice and offer very similar plantar pressure reductions to non-removable offloading devices [14]. However, although removable offloading devices offer similar plantar pressure reductions, they have been found to be much less effective at healing DFU due to people’s low adherence to using these removable devices [8,15,16]. Thus, adherence to using removable offloading devices has been highlighted in the latest international DFU guidelines as an important topic for future research [7].

Guidelines recommend using objective measures to assess adherence to offloading treatment in patients with DFU [7,17]. Examples of such measures include wearable activity monitors to measure the proportion of adherence to total weight-bearing activity [17,18], or temperature monitors to measure the proportion of adherence of total treatment time [19]. However, there are barriers to using these objective monitors, such as costs of monitors [20], privacy concerns [20,21], technical skills required to use [15,22], and limited battery power to use over long periods [23]. On the other hand, people’s self-reported adherence is more commonly used in clinical practice and research, as it has been shown to be quick, affordable, and easy to implement [24]. However, self-reported adherence measures have been shown to be unreliable compared to objectively measured adherence in other similar treatments, such as brace treatments for clubfoot or scoliosis [25,26,27].

To our knowledge, people’s self-reported adherence to offloading treatment among people with DFU has not been tested for validity or reliability. The assessment of validity and reliability of self-reported scales of offloading adherence should guide researchers to know whether they can rely on using self-reporting measures and potentially overcome the aforementioned challenges associated with objective measures of adherence. Furthermore, clinicians should be informed if they can use their patients’ self-reports of adherence as a valid source of assessing adherence to prescribed offloading treatments in clinical practice. Thus, this study aimed to assess the validity and reliability of self-reported adherence to using removable cast walker (RCW) offloading among people with DFUs.

## 2. Materials and Methods

### 2.1. Study Design and Settings

This was a secondary analysis of a larger multi-centre study with a cross-sectional design that aimed to examine adherence to wearing removable cast walkers (RCWs) [16]. Data were collected from three main referral diabetes-related foot clinics in Amman, Jordan, including (i) the National Centre for Diabetes, Endocrinology, and Genetics; (ii) Jordanian Royal Medical Services; and (iii) the Prince Hamza Hospital.

### 2.2. Participants

Eligible participants were adults (>18 years) with diabetes (type 1 or 2) and a plantar DFU, and had been using an RCW for at least four weeks prior to recruitment. A plantar DFU was defined as a full-thickness wound on the plantar surface of the foot in a person with diabetes [16,28]. Removable cast walkers were defined as prefabricated knee-high offloading devices that could be removed by the patient [7,16]. Participants were required to have used RCW for at least four weeks prior, to reduce any effect of initially elevated RCW adherence after receiving offloading treatment as previously identified [29]. Participants who were managed by non-removable offloading devices, unable to ambulate without a walking aid, or had cognitive impairment or illiteracy were excluded [16]. We used data from 53 eligible participants from a larger study who both self-reported adherence and had their adherence objectively measured to test validity [16,30]. The sample size calculation of the larger study was based on five factors, including a final multiple linear regression model with a minimum of 10 participants needed for each included factor and accounting for a 5–10% drop out rate (16). The original sample was considered large enough for this validity study, and a subset of 20 participants were asked to self-report adherence again at a second visit to test reliability based on sample size needed for appropriate interval estimation for Cohen’s kappa [31].

### 2.3. Variables Collected

The definitions for all variables collected have been previously detailed elsewhere [16]. Sociodemographic variables included age, gender, living arrangement, highest education level achieved, employment, and family income (in Jordanian Dinar (JOD)) [16,30]. Medical variables [16,30] included diabetes type, diabetes duration, body mass index (BMI), previous DFU history, current DFU duration, duration of offloading device use [32], peripheral neuropathy [28], peripheral artery disease (PAD) [28], foot deformities, amputations [28,33], DFU area [32], DFU infection, and DFU grade [34].

### 2.4. Outcome Measures

#### 2.4.1. Self-Reported Adherence

Self-reported adherence to using RCW was measured by asking participants to estimate their adherence to using their RCW during total weight-bearing steps on a typical average day, by completing a 10-point visual analogue scale (VAS) converted to a percentage of adherence (0–100%) [30,35,36]. This self-reported adherence measure had been developed, tested for face validity, and translated into the Arabic language as previously detailed elsewhere [30] (see Appendix A for Arabic and English versions of the self-reported adherence measure).

#### 2.4.2. Objective Adherence (Criterion Measure)

Objectively measured adherence to using RCW was measured using a validated dual-activity monitor method over one week [15]. Fitbit Flex© activity monitors were used, which have been shown to be valid and reliable among elderly populations to measure steps [37,38,39,40]. One Fitbit monitor was attached to the RCW to measure the steps when the RCW was used, and the other monitor was worn on the wrist by the study participants to measure their total steps [16]. Participants were instructed to wear wrist monitors at all times for the seven-day period, of which they were reminded daily via text or audio messaging [16]. Participants were concealed from the aim of the measurement to avoid biasing their natural adherence behaviour [17], and otherwise, they received usual adherence instructions from their treating clinicians [16]. After 1 week, monitors were returned, and the steps data were synchronised into 15 min activity units. Adherence to an activity unit was deemed when the activity monitor attached to the RCW recorded at least 50% of the steps recorded by the wrist monitor in the 15 min activity unit [15,16,17,41]. Objective adherence was then reported as the percentage of adherent activity units in the total activity units [15,16,41].

### 2.5. Procedure

At the initial study visit (baseline), all demographic and medical variables were collected, along with participants’ self-reported adherence. The activity monitors were then installed on participants to objectively measure adherence for the one-week period. The activity monitors were returned after one week at a second study visit during regular wound care follow up. At this second visit, the subset of 20 participants was selected by asking every 3rd participant to again self-report their adherence by completing the same aforementioned scale [16,30].

### 2.6. Statistical Analysis

Data analysis was conducted using SPSS 23.0 for Windows (IBM Corp, Armonk, NY, USA). Descriptive analysis included frequencies (proportions), mean (standard deviation (SD)), and median (interquartile range (IQR)). Wilcoxon signed ranks test was used to test the difference between nonparametric adherence outcomes. Pearson’s correlation (r) was used to test the strength of agreement for validity between self-reported adherence and objectively measured adherence outcomes, in which r > 0.75 was considered excellent; r = 0.50–0.75 was good; r = 0.25–0.49 was fair; and r < 0.25 was no agreement [42]. Bland–Altman plots were also used for validity to estimate if the differences between both measurements of adherence led to large mean and 95% limits of agreement and if there was any estimation bias during reporting different levels of adherence. Linear regression was further used to test the significance of the potential proportional bias between the mean difference between self-reported adherence and objective adherence and the mean of these two measurements [43]. Cohen’s kappa was used to reflect the self-reported test–retest reliability agreement in the subset. The Kappa values were considered no agreement in the range 0–0.20; minimal agreement 0.21–0.39; weak agreement 0.40–0.59; moderate agreement 0.60–0.79; strong agreement 0.80–0.90; and almost perfect agreement >0.90 [44].

## 3. Results

### 3.1. Characteristics

Table 1 displays the sociodemographic and medical characteristics of the 53 participants, including a mean (SD) age of 55 (10) years, 77% were male, 94% had type 2 diabetes, 91% had peripheral neuropathy, 26% had PAD, 30% had minor amputation(s), 51% had infected DFU, and 42% had deep DFU, and there was a median (IQR) duration of prior RCW use of 12 (4–32) weeks. Of the 20 participants selected in the subset and asked to test–retest self-reported adherence, 19 completed the self-reported adherence scale at the second visit (retest) and one missed completing the scale.

### 3.2. Adherence

Participants’ median (IQR) self-reported adherence at a baseline of 90% (60–100) was significantly higher than the objectively measured adherence of 35% (19–48) (z = 6.19, *p* < 001). There was no statistical difference between the median (IQR) self-reported adherence reported at the baseline visit (90% (60–100)) and the second visit in the subset of *n* = 19 participants (80% (70–100)) (z = −0.26, *p* = 0.80).

### 3.3. Validity

The self-reported adherence demonstrated only fair agreement with objectively measured adherence (r = 0.46; *p* < 0.05). Figure 1 displays a lower right skewing of self-reported adherence data when plotted against the objective adherence data, indicating an overestimation of self-reported adherence. Figure 2 displays a mean difference (95% limits of agreement) of 43.0% (3.6–89.6) between the self-reported and the objective adherence in the Bland–Altman plots, indicating large differences between self-reported and objective adherence. The linear regression model also identified a significant proportional bias for self-reported adherence (β = 0.46, *p* < 0.01), indicating a significant and systematic overestimation of the self-reported adherence to using RCW.

### 3.4. Reliability

The self-reported test–retest adherence measured at the first baseline visit and second visit in the subset of participants also demonstrated only minimal agreement (Kappa (SE) = 0.36 (0.12); *p* < 0.01).

## 4. Discussion

We tested the validity and reliability of self-reported adherence to using removable offloading treatment (RCW) among people with DFUs. We found a median self-reported adherence of 90% of daily steps, which was a significant overestimation in comparison to the 35% objectively measured adherence using dual-activity monitors. For validity, we found only fair agreement between self-reported adherence and objective measures and noticed a systematic bias towards higher self-reporting of adherence. The bias increased when reporting higher levels of adherence. This indicates that patients with DFUs are not able to accurately assess their offloading adherence. For test–retest reliability, we found minimal agreement between two self-reported adherence measures one week apart, indicating limited stability when patients self-report adherence. These findings suggest that current methods of self-reporting of offloading adherence by patients with DFUs have limited validity and reliability.

Our findings of a significant overestimation of adherence aligns with previous adherence studies in other conditions that compared self-reported and objective measurements, such as in clubfoot, cystic fibrosis, idiopathic scoliosis, and chronic knee pain [25,26,27,45,46]. Studies suggest that patients may overestimate their adherence to avoid conflicts with their clinicians [47], to be socially desirable (i.e., people like to be seen as ”good people”) [48], due to memory bias [24], or due to a distorted perception of adherence itself [49]. In the context of adherence to offloading treatment in people with DFU, a recent qualitative investigation by our group found that patients may overestimate their adherence due to a distorted perception of offloading adherence [30]. Patients considered that adherence to using offloading treatment was only required for weight-bearing activity outside the house or during the daytime, whereas they perceived using their offloading devices inside the house or during night-time when weight-bearing as not being part of adherence requirements [30]. Moreover, a recent qualitative meta-analysis demonstrated that patients with DFUs might not have adequate understandings regarding causation, timelines, and related consequences of DFUs, and this may also lead to a distorted perception of adherence, which may result in inaccurate estimation [50]. We recommend clinicians clarify with patients that total adherence means using their offloading treatment during any weight-bearing activity, outside or inside the house, especially as studies have shown that most weight-bearing activity in this population is performed inside the house [51]. However, while this hypothesis is plausible, further studies are needed to confirm if patients do have a distorted perception of adherence to using offloading or if other factors may be impacting their significant overestimation of adherence to offloading treatment.

Overall, estimating self-reported adherence to using the offloading device may be a challenging task for patients with DFU. The significant overestimation by patients found in this study is likely to mislead patients and clinicians as to the effectiveness of removable offloading treatments to heal DFUs. Therefore, when measuring adherence, we recommend clinicians and researchers do not use the current self-reported measures, and instead use recommended objective measures of offloading adherence [17,23], such as dual-activity monitors to measure adherence during weight-bearing activity [20] or the in-device temperature monitors to measure adherence during treatment time [19]. Furthermore, the recent incorporation of objective self-monitoring adherence technology within offloading devices (“smart offloading boots”) seems a promising option to provide patients and clinicians with objective real-time monitoring of adherence in the future [18]. This may result in adherence enforcement to using offloading treatment, which may positively contribute to enhanced adherence and in turn improved healing rates similar to those found in non-removable offloading devices. Such self-monitoring technology may also help researchers to examine other factors associated with adherence to offloading treatment among patients with DFUs. Otherwise, developing demonstrated valid and reliable self-reported measures of adherence to offloading treatment in the future would still be a valuable addition to the field, as self-reported measures have been found quick, affordable, and easy to implement.

### Strengths and Limitations

The strengths of this study included comparing a self-reported method of reporting adherence typically used in clinical practice and research against a recommended validated objective dual activity monitor method in an appropriate sample to test validity and reliability. In addition, we used a minimum of four weeks’ prior experience of using RCW as an inclusion criterion to be more representative of the typical adherence behaviour patterns of patients with DFU [29]. However, the limitations of this study also need to be considered. First, there was a slight difference in the units of measurement used for the self-reported and objective adherence measures. The self-reported measure used the percentage of daily steps, while the objective measure used the percentage of daily activity units as recommended for objective adherence measures [15,17]. However, we consider that these slight differences in units should not have had any major impact on findings, as activity units are also made up of the percentage of steps using the offloading device per 15 min period, and as such, are a similar measure to the percentage of daily steps. Furthermore, it would be challenging for patients to understand and self-report adherence for daily activity units. Second, participants were asked to self-report their adherence for an average typical day, whereas the objective measure captured percentage adherence for a set one-week period. However, we included participants with at least four weeks of prior use of RCW treatment [29], and we chose the one-week period because four weekdays and one weekend day have been shown to be representative of average daily activity in physical activity studies [16,17,52,53,54], and thus we should have a representative sample of the true daily activity of a DFU population. Third, the sample size was based on the sample size calculations of our main study that used regression and not specifically for this analysis of validity and reliability [16]. Sample size calculations and potentially larger sample sizes are recommended for similar studies in the future. Last, there was a possibility of not wearing the wrist activity monitors; however, we attempted to minimise non-adherence of wearing wrist trackers via daily reminders to participants [16,30].

## 5. Conclusions

This study suggests that people with DFUs substantially overestimate their adherence to using offloading treatment compared to objective measures. In addition, we found only fair agreement at best to support the validity and reliability of self-reported adherence and significant proportional bias. Thus, it seems that self-reported adherence is not an accurate or reliable measure of actual adherence. We recommend that clinicians and researchers adopt objective measures to avoid misleading measurements of removable offloading adherence until improved self-reported measures are developed and tested in the future.

## Figures and Tables

**Figure 1 sensors-23-04423-f001:**
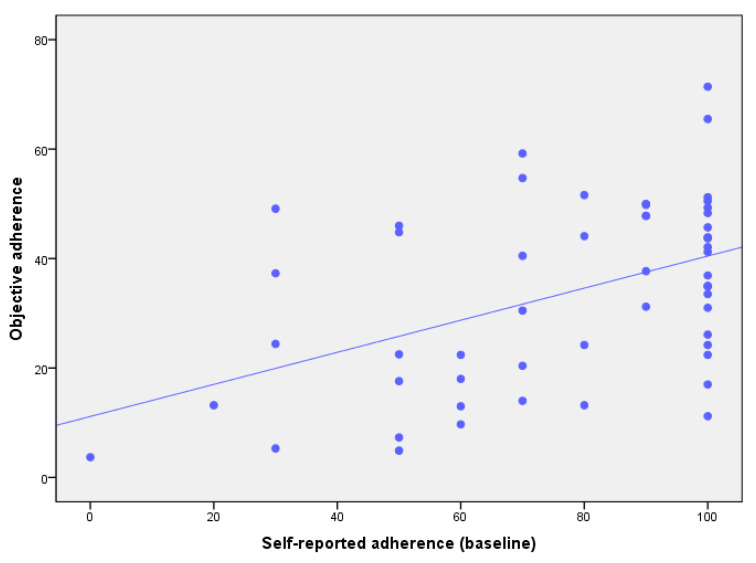
Scatter plots depicting self-reported adherence percentage compared to the objectively measured adherence percentage to using RCW.

**Figure 2 sensors-23-04423-f002:**
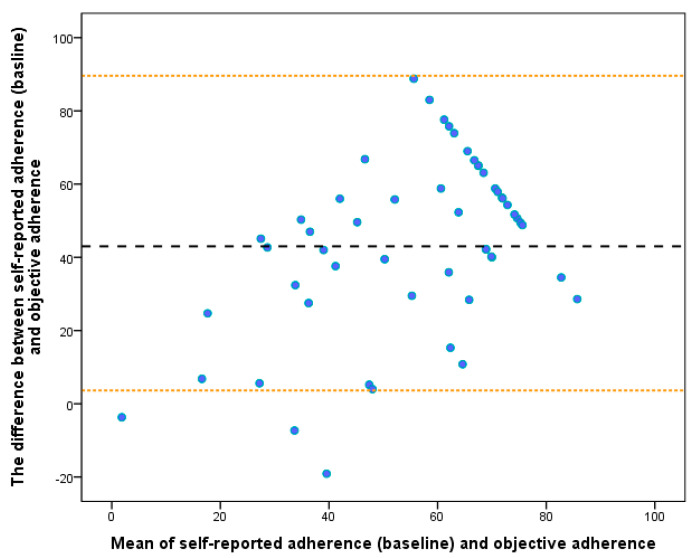
Bland–Altman plot depicting the agreement between the self-reported and objectively measured adherence to wearing RCW using activity monitors. The dashed black line represents the mean difference between self-reported and objective adherence, and the orange dashed lines represent the upper and lower limits of agreement.

**Table 1 sensors-23-04423-t001:** Participant sociodemographic and medical characteristics (number (%) or mean ± SD unless otherwise stated^).

Characteristics	Total
Numbers	53
Age (years)	55.3 (9.9)
Males	41 (77.4%)
Living with family	49 (92.5%)
Secondary school education	24 (45.3%)
Retired	17 (32.1%)
Family income (JD) ^	400 (300–712.5)
Type 2 DM	50 (94.3%)
Duration of diabetes (years)	17.7 (7.0)
HbA1c (%, mmol/L)	8.9 (2.1)
BMI	31 (6.5)
Daily steps (wrist activity monitor) ^	2758.4 (1729–4676)
Neuropathy	48 (90.6%)
PAD	14 (26.4%)
Foot deformities	38 (71.7%)
Minor amputations	16 (30.2%)
Major amputations	0 (0%)
History of previous ulceration	35 (67.3%)
Duration of ulcer (weeks) ^	16 (5.0–38)
Ulcer size (cm^2^) ^	1.5 (0.5–6.0)
Deep ulcer (UTWCS Grade 2 or 3)	22 (41.5%)
Ulcer infection	27 (50.9%)
Duration of RCW (weeks) ^	12 (4.0–32.0)

^ Displayed as median (IQR). Abbreviations: BMI: body mass index, CM: centimetre, DM: diabetes mellitus, JD: Jordanian Dinar (JOD), PAD: peripheral artery disease, RCW: removable cast walker, SD: standard deviation, UTWCS: University of Texas Wound Classification System.

## Data Availability

Data available on request due to privacy restrictions. The data presented in this study are available on request from the corresponding author.

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
