# Peer review of "The Validity and Reliability of Self-Reported Adherence to Using Offloading Treatment in People with Diabetes-Related Foot Ulcers"

_sensors, 2023, doi:10.3390/s23094423_

Round 1

Reviewer 1 Report

1) row 54, the removable offloading devices offer a similar plantar pressure reduction regardless removable cast type, but was lower then reported for non-removable  offloading devices.

2) row 160, specify number and percentage  for characteristics of diabetic comorbidity: only peripheral neuropathy, only PAD and neuroischemic

3)row 160: specify what grade of infection according to IWGDF classification (mild/moderate/severe) 

Reviewer 2 Report

First of all, I appreciate the work done by the team.
Regarding the methodology, the time of presence of the wound seems too long.
It is very likely that patients with younger lesions have greater adherence than older ones.
It also seems to me that the size of the lesion, depth and the presence or absence of infection should influence adherence due to the sensation of severity of the wound for the patient.
It would be interesting to know if any comparison of results could be made taking into account the sensation of severity and the time of evolution of the wound.
This is an interesting study, but its interest would increase taking into account the confounding variables mentioned.
King Regards

Reviewer 3 Report

Dear Author's

Thank you for the opportunity to familiarize yourself with the results of the
manuscript. I appreciate the Authors' contribution to its creation. However,
in my opinion as well as the reader, the reviewed work is not an innovative
scientific achievement and does not indicate a creative, original and
innovative research problem. I leave the final decision to the Editor-in-chief.

Reviewer 4 Report

1. Intro can be still be elaborated to assert more on need of the study 

2. In table 1 I see 53 participants but I see only 48 with neuropathy and rest with PAD, is it truly reflecting  a population suffering from the diabetes related complication 

3. figure 1 scatter plot the authors says for 53 participants but actual participants seems to be 49. Can you please recheck and clarify? 

4.  Figure 2 also represents data from 49 individuals 

5. how was the sample size calculated , kindly clarify 

6. Optional : I feel the authors can add highlights to the study to put there points or can add a graphical abstract 

Round 2

Reviewer 3 Report

Dear Author's 

Thank you for your comment. The final decision is forwarded to the Editor in Chief.

Reviewer 4 Report

1. point 5 and 6 are not addressed appropriately 

If sample size wasnt calculated should be listed out under limitation. I am unclear what was the exact formula used , which was the reference study taken into account for the same. 

point 6 Authors should build a graphical abstract so that researchers  from all field can understand the study 

Round 3

Reviewer 4 Report

1. It is important that study references the parent study as the publication should not fall in contradiction to publication policies 

2. Graphical abstract looks good , but I would suggest to also add inference for readers, this good reliability means what for clinicians and patient population. 
